# Cohen’s Kappa Coefficient as a Measure to Assess Classification Improvement following the Addition of a New Marker to a Regression Model

**DOI:** 10.3390/ijerph191610213

**Published:** 2022-08-17

**Authors:** Barbara Więckowska, Katarzyna B. Kubiak, Paulina Jóźwiak, Wacław Moryson, Barbara Stawińska-Witoszyńska

**Affiliations:** 1Department of Computer Science and Statistics, Poznan University of Medical Sciences, 60-806 Poznan, Poland; 2Department of Preventive Medicine, Poznan University of Medical Sciences, 60-781 Poznan, Poland; 3Department of Epidemiology and Hygiene, Chair of Social Medicine, Poznan University of Medical Sciences, 60-806 Poznan, Poland

**Keywords:** Cohen’s Kappa coefficient, NRI net reclassification ratio, logistic regression model, reclassification, AUC

## Abstract

The need to search for new measures describing the classification of a logistic regression model stems from the difficulty in searching for previously unknown factors that predict the occurrence of a disease. A classification quality assessment can be performed by testing the change in the area under the receiver operating characteristic curve (AUC). Another approach is to use the Net Reclassification Improvement (NRI), which is based on a comparison between the predicted risk, determined on the basis of the basic model, and the predicted risk that comes from the model enriched with an additional factor. In this paper, we draw attention to Cohen’s Kappa coefficient, which examines the actual agreement in the correction of a random agreement. We proposed to extend this coefficient so that it may be used to detect the quality of a logistic regression model reclassification. The results provided by Kappa‘s reclassification were compared with the results obtained using NRI. The random variables’ distribution attached to the model on the classification change, measured by NRI, Kappa, and AUC, was presented. A simulation study was conducted on the basis of a cohort containing 3971 Poles obtained during the implementation of a lower limb atherosclerosis prevention program.

## 1. Introduction

In predictive regression models based on known factors identifying a new yet-unknown disease predictor is a difficult task, due to the fact that many factors have already been discovered. Therefore, there is a need for research on new measures describing classification changes once a new factor has been added to the model [1]. The relationship between *p*-values and hypothesis inference is necessary but not sufficient. This is one of the key themes of the American Statistical Association’s report on statistical significance and *p*-values [2]. Statistical evaluations are paying increasing attention to effect sizes. Studies comparing other metrics to measure the significance and robustness of hypotheses are helping to improve decision-making [3,4,5,6,7,8]. Some editors have even banned the use of the *p*-values (testing the significance of the null hypothesis) in their journals [9].

The study of c statistic, or field size (AUC) under the ROC curve, has a long history of over forty years of use in many areas. It is a measure commonly used in medicine as well. It works on continuous variables and is used, among others, to assess clinical suitability in both diagnostic models, such as logistic regression models, and prognostic models with a time dimension, such as survival models.

We can also assess the quality of diagnostics or model prediction based on a discreet popular Kappa Cohen coefficient [10,11]. Many combinations and generalizations of Kappa, such as stratified Kappa [12] and weighted Kappa [13]. Kappa estimated on the basis of a large volume of stratified and unbalanced data [14,15,16] and Kappa modelled with accompanying effects in GEE regression models [17,18] have been put forward in literature. A guide to the different versions of Kappa applied in medical analyses is provided by Helena Chmura Kraemer et al. 2002 [19]. Both these measures, AUC and Kappa, assessed the quality of regression model classification.

The inclusion of further risk factors and subsequent markers in regression models improves future disease prediction or diagnosis and is seen in the models as an increase in AUC. Determining the change in the c statistic after adding new information to the model in the form of new potential risk factors is a commonly used mechanism to determine how the change in model classification quality. However, difficulties with interpretation have long been raised. These difficulties usually relate to small, though statistically significant, changes in the statistics, and to the fact that the scale of improvement is related to the efficiency of the basic model. That is, given the same risk factor or marker, greater field improvement is possible under conditions where the base regression model exhibits less discrimination (low accuracy) compared to settings where it exhibits more discrimination (high accuracy) [20,21,22]. In the search for a better criterion for assessing model classification improvements, attention was paid to reclassification. In general, reclassification entails comparing the estimated risk obtained for each patient based on two regression models: the basic and the one with a new marker added to check whether its reclassification occurred up (higher risk of an event) or down (lower risk of an event). For this reason, the NRI (Net Reclassification Improvement), introduced and developed by Michael J. Pencina [23,24,25], is gaining popularity. NRI can be used to check the size of a classification change after adding a new marker to an existing model that may change the clinical management of such a patient. However, despite the high potential of this measure, it has not replaced ROC. The reason for this is evident in the doubts raised regarding the correctness of some calculations and interpretations that have appeared in statistics journals [26] and in reputable medical journals [24,27,28,29,30,31,32]. Many misunderstandings stem from failure to use the coefficient in accordance with the author’s assumptions and recommendations [21,23,24,33]. In cardiology, where this factor is most widely used, an attempt was made to summarize strengths and weaknesses and provide guidelines for its safe use [34]. By following these recommendations, it is possible to eliminate most errors safely. However, a major disadvantage of NRI is that it gives a beneficial up and down reclassification percentage, even if a random variable has been added to the model. That is why we have focused on improving this flaw. Cohen’s Kappa coefficient seems to be a good candidate due to the fact that it examines the actual agreement of the measurement, with a correction for random agreement. The classic use of the Kappa coefficient entails testing the compatibility of two measurements. For regression models, it is the agreement of the obtained forecast with the actual classification of patients. However, when adding subsequent information about the same research subjects to the model, there is a need to compare the classification quality of the two models. The first of the suggested solutions is to use the test to compare the two correlated Kappa results proposed by Hongyuan Cao et al. (2016) [35]. Nonetheless, this test is based on the determination of conditional coefficients, not a reclassification of each patient.

The aim of this work is to extend the Cohen Kappa coefficient so that it can be used to check the reclassification quality in a regression model and to compare the results described by the Cohen Kappa coefficient with continuous, categorical, and unit NRI results.

## 2. Materials and Methods

Ordinarily, adding a new variable to a model does not leave reclassification unchanged. If a factor added to a model is completely random, the reclassification may slightly deteriorate or slightly improve. However, if the new variable is an effective classification factor, its addition should improve reclassification and the magnitude of this improvement should far outweigh any possible random improvements.

Table 1 shows the results of reclassifying a group of *n* patients into two groups, i.e., a group of people diagnosed with a disease and a group of people who were not diagnosed with a disease, and the corresponding expected numbers.

Let us denote the number of patients correctly reclassified as disease-free by *O′* = *a_O_* − *e_O_* and the number of patients correctly reclassified as diseased by *O″* = *f_O_* − *b_O_*.

Thus, *NRI* for downward and upward reclassifications is:$$NRI_{down}=\frac{O^{\prime }}{\#disease-free},\;NRI_{up}=\frac{O^{\prime \prime }}{\;\#diseased}$$

Total *NRI* is: $NRI=NRI_{down}+NRI_{up}$.

The study on improving classification using *NRI* takes into account *O′* and *O″*, i.e., the number of correctly reclassified disease-free and diseased people, ignoring the part of this change that may be the result of a random impact. The actual number of reclassifications should be reduced by the number of reclassifications that could arise when adding a random variable to the model, i.e., the value *E′* = *a_E_* − *e_E_* and *E″* = *f_E_* − *b_E_*. The random number of reclassifications concerns disease-free (*E′*) and diseased people (*E″*). Thus, the number of people for whom we can expect a real improvement in reclassification, i.e., non-random, is *O′* − *E′* disease-free and *O″* − *E″* diseased, respectively.

If by *E* = (*a_E_* + *f_E_*) we denote the random (expected) number of correct reclassifications, and by *O* = (*a_O_* + *f_O_*) the observed number of correct reclassifications, then the calculations can be reduced to determining the Cohen’s Kappa compliance coefficient. However, a symmetric table is needed to determine the Kappa coefficient. After adding a new variable to the basic model, the change in the predicted probability may have three variants, i.e., reclassification down, up, and no reclassification, and the reality is a binary variable with two categories, i.e., disease-free and diseased. In order to maintain symmetry of the table, the addition of a hidden category is necessary. No examined person can belong to the hidden category, therefore the counts in this category will be zero (Table 2).

We calculate the Kappa coefficient based on the diagonal of the matrix as the observed number of correct reclassifications *O* = (*a_O_* + *f_O_*) reduced by the expected number of correct reclassifications *E* = (*a_E_* + *f_E_*).
$$\kappa =\frac{O-E}{n-E}$$

Thus, the Kappa coefficient yields the agreement measure of up or down reclassifications with the actual occurrence or absence of disease, i.e., among people who may have been reclassified in a non-random way (denominator) and how many have actually been reclassified in a non-random way (numerator).

Interpretation of Cohen’s Kappa reclassification:*κ* > 0 indicates better reclassification after adding a new variable,*κ* < 0 indicates worse reclassification after adding a new variable,*κ* = 0 indicates that there are no changes in the reclassification.

In addition, we note that it may be presented in a table showing changes in classification after adding a new variable: *O* − *E* = *O*′ − *E*′ = *O*″ − *E*″ = *x*
*O* − *E* = (*a_O_* + *f_O_*) − (*a_E_* + *f_E_*) = *x*
*O*′ − *E*′= (*a_O_* − *e_O_*) − (*a_E_* − *e_E_*) = *x*
*O*″ − *E*″ = (*f_O_* − *b_O_*) − (*f_E_* − *b_E_*) = *x*

The *x*–value shows the number of correct reclassifications reduced by the random number of correct reclassifications.

### 2.1. Example

To illustrate how to determine and interpret the number of correct reclassifications *x* and Cohen’s Kappa reclassifications as well as the known NRI, consider a 50-element sample. An example of the number of reclassifications after adding a new variable and the number of random reclassifications for this variable is presented in Table 3.

The Net Reclassification Improvement was calculated based only on the observed frequencies as:NRI_Down_ = 0.50, NRI_Up_ = 0.25, NRI = 0.75

We calculate the number of correct reclassifications *x,* taking into account the expected frequencies:*x* = *O* − *E* = (*a_O_* + *f_O_*) − (*a_E_* + *f_E_*) = *a_O_* − *a_E_* + *f_O_* − *f_E_* = 9
*x* = *O*′ − *E*′ = (*a_O_* − *e_O_*) − (*a_E_* − *e_E_*) = 9
*x* = *O*″ − *E*″ = (*f_O_* − *b_O_*) − (*f_E_* − *b_E_*) = 9

Thus, we can say that nine people (18% of the study group) are better diagnosed than if the added variable was random. An example of a graphical interpretation is shown in Figure 1.

With the obtained reclassification, more people saw a decrease than an increase in categorization, i.e., a decrease was observed in *a*_0_ + *b*_0_ = 25 people, i.e., 50% of the study group, while an increase in *e*_0_ + *f*_0_ = 15 people (30% of the analyzed group). It is known that by adding this variable we can improve the forecast for disease-free individuals more than for the diseased. We expected that in the groups of the diseased and disease-free, the probability would decrease as described above, i.e., in half of the subjects in the group of thirty healthy people (*a_E_* = 15) and in half of the subjects in the twenty diseased (*b_E_* = 10). Meanwhile, the reclassification turned out to be more favorable than expected and the probability of disease decreased more in healthy people (*a_O_* = 20 people, i.e., 66.7%). For those diagnosed with a disease, the decrease concerned *b_O_* = 5 people, i.e., 25% of this group. Similarly, we expected that in the groups of the diseased and disease-free the probability would increase as described above, i.e., 30% of the subjects in the healthy group (*e_E_* = 9) and in in the diseased group (*f_E_* = 6). Meanwhile, again, the reclassification turned out to be more favorable, i.e., the probability of disease increased more in diseased people (*f_O_* = 10, 50%) and less in the healthy group (*b_O_* = 5, 16.7%). 

We compare this reference to expected values by applying the Kappa Cohen:$$k=\frac{O-E}{n-E}=\frac{x}{n-E}=\frac{9}{50-\left(15+6\right)}=\frac{9}{29}=0.31$$

We know that some people will undergo random and correct reclassification (*E* = 21), hence the remaining number of people who may be subject to a non-random reclassification decreased to 29 (Kappa denominator). Therefore, interpreting Kappa, we can say that these 9 people (Kappa numerator) constitute 31% of those who can be reclassified in a non-random way, i.e., unexpected (*n* − *E*).

Because the value of *x* is based on the difference in observed and expected frequencies, and expected frequencies are often real numbers, there can be a problem with rounding *x*. We propose rounding the number of individuals to the nearest whole number in accordance with accepted rounding rules.

### 2.2. Plan of Simulation Study

Cardiovascular diseases remain the leading cause of death in the world. It is commonly assumed that age, gender, high blood pressure, high blood cholesterol, smoking, obesity, and diabetes are responsible for causing them.

The overarching goal of CVD prevention is to reduce morbidity and mortality through actions at a population level or unit oriented to promote a healthy lifestyle, early identification of patients at risk of CVD, and reduction of risk factors.

Furthermore, cardiovascular risk assessment is important for medical professionals. It may help them guide the intensity of treatment and predict cardiovascular risk in individual patients. The risk scale Framingham (in the United States) and the SCORE system (in Europe) is most often used in clinical practice.

The first is a gender-specific algorithm used to estimate the 10-year cardiovascular risk of an individual based on the following risk factors: dyslipidaemia, age range, hypertension treatment, smoking, and total cholesterol [35,36]. The SCORE system uses similar risk factors such as: gender, age, smoking, systolic blood pressure, total cholesterol, and estimates fatal cardiovascular disease events over a ten-year period [37,38]. These scales are intended for primary prevention in patients without previous cardiovascular incidents.

In the study, the authors used data obtained during the implementation of the prevention program of arteriosclerosis of the lower extremities financed by the Ministry of Health in Poland under the POLKARD program.

The population participating in the program was selected at random and was representative of the Polish population in terms of gender and age. The authors studied 3971 patients, whose data were valid and fully available.

Based on a score table developed by Boston University together with the National Heart, Lung, and Blood Institute for the Framingham Heart Study, CVD risk was estimated for both women and men. For the purposes of the study, patient results were divided into two groups according to the estimated risk of developing cardiovascular disease.

The first group comprised 1765 patients with an estimated risk ≥20% (CVD risk group) and there were 2206 patients with an estimated risk <20% (healthy group) in the second group.

The CVD risk variable was a dependent variable, forecasted based on a logistic regression model.

For the purpose of this paper, we needed to build a base model and extended models. In a nutshell, the algorithm for selecting new variables for an already existing model uses a series of repetitive actions. A given base model, Model 1, is extended by one variable which creates a series of new Models 1 from which the best one is selected—Model 2 (Model 1 + one variable). Then, the algorithm repeats the same procedure using the Model 2 (Model 1 + one variable) so many Models 2 expanded by one extra variable are created, from which the best one is selected—Model 3 (Model 2 + one variable). In each step, the model may be corrected by removing an unnecessary variable. However, our aim is not to undergo the whole algorithm of selecting the best CVD prediction model, but to describe one step of such procedure, i.e., creating models extended by one variable, and show how Kappa can be used to evaluate the prediction improvement of such models.

### 2.3. Selection of Candidates for Extended Models

For the construction of extended models, we needed both variables that we knew are strongly associated with CVD risk and those that are not. The CVD risk markers defined that way allow us to check whether both measures, i.e., NRI and Kappa, as expected, respond correctly to the addition of the variables to the base model, i.e., show a significant improvement in prediction when the model is expanded with variables strongly associated with CVD, and no such improvement when the model is expanded with random variables.

We had at our disposal 12 variables collected in the POLCARD study: gender, age, smoking, systolic blood pressure, total cholesterol, and estimates fatal cardiovascular disease events over a ten-year period, BMI, place of residence, marital status, income, daily activity, and education. The first six were converted into SCORE according to the guidelines [39]. As a result, 7 variables remained at our disposal for analysis, i.e., BMI, place of residence, marital status, income, daily activity, education, and the SCORE, indirectly taking into account the information carried by all the variables from which it was constructed. From these seven variables, we decided to select two variables which were most suited for the construction of extended models, i.e., variables with the highest R2 and exceeding the threshold of statistical significance according to the Wald test in one-dimensional logistic regression models. In the analyses, as candidates for new CVD markers, 6 simulated variables were also included. They were randomly selected, so they should not be related to the dependent variable. The random selection was conducted according to the uniform distribution in the range from 0 to 100, normal distribution (0,1), Poisson distribution (λ = 4), exponential distribution (λ = 1), bimodal distribution (probability of success *p* = 0.1), and bimodal distribution (probability of success *p* = 0.5).

### 2.4. Construction of the Basic Model

Of the seven POLCARD study variables, two could not be included in the base model because they were selected as candidates for extended models. The remaining five variables were subjected to a forward stepwise regression procedure to eventually create a smaller baseline model devoid of irrelevant variables.

Each variable involved in the analysis was described with an odds ratio, along with a 95% confidence interval and a Wald test *p*-value. For models with an additional marker, results of the Wald test examining the significance of a given variable, results of the likelihood ratio test for comparing the basic model with the extended model, area under the ROC curve and statistical significance of the field change after adding a new variable were given. The quality of the reclassification was tested using the Kappa Cohen and the NRI coefficient. The analysis was performed for continuous risk change, 1% risk change, and categorial risk change based on the percentage of the CVD risk group, which in our case was 1765/3971 = 0.4444. The PQStat v1.6.8 software (Poznań, Poland) was used for the calculations. A significance level of 0.05 was assumed.

## 3. Results

One-dimensional logistic regression models indicated BMI, income, daily activity, education, and SCORE as variables significantly related to CVD risk (Table 4). The high SCORE obtained the highest odds ratio of OR [95%CI] = 22.66 [18.27; 28.12]. Random variables did not exceed the statistical significance threshold except for the variable with a Poisson distribution, for which *p* = 0.0443. SCORE and education were selected as candidates for the extended models as they were variables that exceeded the threshold for the statistical significance in unidimensional analyses and obtained the largest R^2^ (Nagelkerke) values. In the baseline model, after applying forward stepwise regression, three variables remained: BMI, income, and daily activity.

The basic logistic regression model was built based on variables such as BMI, income, and daily activity, which showed statistical significance in one-dimensional analysis (Table 5). The quality of the basic model classification described by the ROC curve, although statistically significant (*p* < 0.0001), is still small AUC = 0.59 [0.57, 0.60]. However, it still appears feasible to find variables that would significantly improve the quality of classification.

Statistical significance determined by the Wald test indicates that only education and SCORE are factors associated with CVD risk regardless of the variables in the basic model, i.e., BMI, income, and daily activity (*p* < 0.0001). The *p*-value obtained for a variable with Poisson distribution is also noteworthy. Despite independent sampling of data from the Poisson distribution, a *p*-value of the Wald test is close to statistical significance (*p* = 0.0550). Statistical significance after adding the education and SCORE variable, as well as the low *p*-value for a variable with a Poisson distribution (*p* = 0.0549), is still valid for the likelihood ratio test. The problem with the variable with the Poisson distribution disappears in ROC curve comparison (*p* = 0.3206).

For a continuous change of risk (Table 6), i.e., by any small amount, 311 people, which represents 7.82% of all respondents, were better reclassified after adding education than after adding a random variable. Kappa value is κ [95%CI] = 0.16 [0.13, 0.19] and NRI [95%CI] = 0.32 [0.26, 0.38]. After the inclusion of SCORE in the model, original classifications were changed for 1035 (26.06%) people to the correct ones, which increased the classification accuracy of the model. In this case, Kappa value is κ [95%CI] = 0.5 [0.48, 0.53] and NRI [95%CI] = 1.06 [1.01, 1.10]. The statistical significance of the Kappa coefficient is very similar to the significance of the NRI. Only variables that represent education and SCORE improved the prediction of the basic model significantly (*p* < 0.0001). However, adding a random variable with a Poisson distribution also drastically reduces the *p*-value (*p* ≈ 0.09).

Solving the problem of accidental statistical significance for Kappa and for NRI, after adding random variables such as a Poisson distribution variable, seems to solve the step increase for CVD risk. Table 7 shows the results obtained for Kappa and NRI when the change in CVD risk is counted only when the up or down step change is greater than 1% (0.01). We called these measures unit-Kappa unit-NRI. For the model extended by the variable with a Poisson distribution, *p*-value ≈ 0.16, the value of the Kappa coefficient and NRI decreased. For education and SCORE, Kappa and NRI obtained almost the same values as in the continuous models, which do not assume a minimal step. For education unit-κ [95%CI] = 0.15 [0.12, 0.18], unit-NRI [95%CI] = 0.31 [0.26, 0.38], for SCORE unit-κ [95%CI] = 0.50 [0.48, 0.53], unit-NRI [95%CI] = 1.05 [1.01, 1.10]

For a reclassification change with respect to the set cut-off point of *p* = 0.4444, i.e., the calculation of the categorical Kappa (*p*) and the categorical NRI (*p*), similar results for these two measures were obtained (Table 8). Again, adding education and SCORE significantly improves the prediction of the basic model (*p* < 0.0001), and the test probability value for a variable with a Poisson distribution is at a safe level of *p*-value ≈ 0.29. The number of correct reclassifications for education is now 52 (1.3%) and for SCORE 397 (10%). For education κ(*p*) [95%CI] = 0.10 [0.01, 0.02], NRI (*p*) [95%CI] = 0.06 [0.03, 0.09], for SCORE κ (*p*) [95%CI] = 0.13 [0.11, 0.14], NRI (*p*) [95%CI] = 0.40 [0.37, 0.44].

## 4. Discussion

Adding a new random variable to the regression model improves model fit, even if this variable is not significantly associated with the dependent variable. It can be expected that adding a random variable to the model will also have an impact on the quality of reclassification. Therefore, it is important to indicate how the quality of the reclassification changes after excluding possible random reclassification. The NRI does not include the required correction, so we proposed using the Cohen Kappa coefficient for this purpose. We noticed that this indicator works based on the number of people correctly reclassified, minus the number of people who can be correctly randomly reclassified. A desirable solution seems to be quoting this corrected number of properly classified people next to the Kappa value in medical works, as well as presenting a graphical interpretation of the process of its determination, which we presented in this work.

To show the advantages and disadvantages of the Kappa coefficient proposed for reclassification, we presented its results obtained on a large data set describing the risk of CVD. The first reference point for the conducted tests were the results obtained for the ROC curve and the likelihood ratio test [31,32,40]. The likelihood ratio test is recommended for use in models in which reclassification quality is measured using NRI. The area under the ROC curve in the base model was AUC [95%CI] = 0.59 [0.57, 0.6]. After adding an independent random variable with a given distribution to the model, the AUC did not change visibly for any of the simulated distributions. Statistically significant changes (*p* < 0.0001) were only noted after adding variables whose impact on CVD risk was expected, i.e., education and SCORE. The likelihood ratio test, calculated for the same models, obtained results consistent with those obtained on the basis of the ROC curve. Unfortunately, it should be noted that, although the likelihood ratio test is recommended for use alongside the NRI index, after adding a random variable with a Poisson distribution to the basic model, similarly to the Wald test, it yielded a *p*-value dangerously close to the statistical significance threshold (*p* = 0.0549). Therefore, we considered further results obtained for Kappa and NRI, keeping in mind this particular random variable with a Poisson distribution, as well as education and SCORE.

Analyzing the risk value from the regression model as a continuous variable, we noticed that the Kappa coefficient was about two times less than the continuous NRI proposed by Pencina [24]. Kappa is based on the observed number of correct reclassifications reduced by the random number of correct reclassifications, hence it was expected to obtain a lower Kappa value relative to NRI. The addition of random variables to the model resulted in these measures being close to zero. Similarly, the AUC size changed only slightly. Statistical significance was only achieved after adding education or SCORE. Unfortunately, the low *p*-value for NRI and for Kappa (*p* ≈ 0.09), obtained after taking into account the Poisson distribution variable, may be a cause for concern. It seems to confirm the fears of many researchers regarding the use of this measure [24,26,27,28,29,30,31,32]. When considering continuous risk, the excessive sensitivity of the model may be due to the fact that we counted even rather minor changes in the size of the risk. Consequently, continuous NRI is often positive for relatively weak markers [21]. For this reason, we suggested that both NRI and Kappa be considered in terms of unit changes, i.e., unit-NRI and unit-Kappa. As a result, we achieved significant improvement. A larger step, i.e., exceeding 1% risk, slightly reduced both coefficients. It contributed to a favorable change in the *p*-value of tests of their statistical significance. Pencina advised against comparing continuous NRI with categorical NRI [21]. We also recommended a comparison of unit-NRI and unit-Kappa with the Kappa and NRI that measured unit risk change obtained in other studies and not with other types of these measures, i.e., based on different risk-sharing.

An issue that we did not undertake in this study, and which seems to be worth considering, is the multi-unit NRI and multi-unit Kappa. For such measures, we can assume that the patient has a change in risk if it is not just greater by one unit (1%) but by a larger amount. The weakness of continuous NRI is that most changes in the predicted risk do not translate into changes in clinical management, i.e., people whose risk changes by 0.5% are unlikely to be treated differently. We will not obtain another approach using the unit change under consideration here, i.e., 1%. Nonetheless, a person who has a 10% increase or decrease in risk may be of interest. Moreover, a corresponding increase in the step leads to the determination of measures with similar advantages to clinical NRI. Clinical NRI is a measure that determines reclassification for a particular, specific risk category. Particularly important from a clinical point of view is the category with high uncertainty of classification, i.e., those with a medium risk of, among others, coronary artery disease [41,42,43]. The use of NRI or Kappa with a large step, e.g., 33.33%, artificially creates an area for three categories, i.e., low risk from 0% to 33.33%, medium from over 33.33% to 66.66%, and high risk from over 66.66% to 100%. These measures examine the possibility of improving the classification only for people in the middle group, as people with risk lower than 33.33% (low-risk group) cannot be reclassified downwards seeing that the step needed for reclassification is 33.33%, and similarly those with a risk higher than 66.66% (i.e., high-risk group) cannot be reclassified upwards. In addition, NRI (33.33) and Kappa (33.33) may gain an advantage over clinical NRI since it is considered biased [44], seeing that it does not take into account the adverse reclassification from extreme risk categories to an intermediate risk category [45]. Thus, correction is required [46]. However, in the example given, reclassification from extreme risk categories to medium categories is possible, which means there is no concern that NRI (33.33) and Kappa (33.33) will be artificially inflated. Therefore, it seems appropriate to suggest that further research should be directed toward this type of multi-unit measures.

Essentially, the basic approach proposed by Pencina to determine NRI is to define risk thresholds, which provides an opportunity to build the categories mentioned earlier [23]. The categorical NRI value has an important advantage over the continuous value. It presents the possibility of arbitrarily setting risk cut-off points, so that the reclassification to a higher risk category will translate into clinical practice. It can be used when the reclassification thresholds are well justified. The ability to set thresholds is, on the one hand, an advantage, but, on the other hand, a disadvantage due to problems with incorrect selection of cut-off points and overestimation (too optimistic result) of this measure. These problems can be avoided by dividing patients into two groups: above and below the event rate [25,29]. The NRI classified by event rate, designated as NRI (*p*), has a number of interesting statistical properties. In particular, NRI (*p*) is resistant to the modelling of incorrect calibration and cannot be fooled by adding random noise [25]. The analysis presented in the study confirmed the resistance of the test examining the significance of NRI (*p*) and Kappa (*p*) to the addition of a random variable with a Poisson distribution, which caused major problems in the analysis based on the continuous risk of CVD. Unfortunately, different reclassification thresholds are used for coefficients based on the event rate, due to the fact that event rates in different populations often vary. For this reason, most forms of NRI are impossible to compare and hence to interpret. The unit or multi-unit NRI and Kappa proposed in this paper are free of such issues.

Furthermore, this paper did not examine how the Kappa designated for reclassification is affected by incorrect calibration, which was noted, among others, by Hilden [27]. By conducting a direct comparison of Framingham’s risk function, i.e., using published coefficients and basic risk, and comparing with the new risk function developed on the basis of own data, an NRI can be found that favors a new model with no difference in AUC [47,48]. For our data, incorrect calibration could not be analyzed because we performed the analysis on the same data set. As a result, we cannot rule out its impact on Kappa results when the analysis is performed on uncalibrated models based on different populations.

## 5. Conclusions

Cohen’s Kappa coefficient, used to measure reclassification in logistic regression models, enables an assessment and interpretation of the reclassification size adjusted for the number of possible random reclassification. The construction of Cohen’s Kappa coefficient, which was used for reclassification testing in logistic regression models, facilitates a graphical presentation of obtaining the number *x*, as well as a description of results in the form of the number of people, i.e., the number of observed correct reclassifications reduced by the random number of correct reclassifications. The significance of Cohen’s Kappa coefficient used to study the reclassification in logistic regression models is similar to the significance of the NRI coefficient. On top of that, the Kappa value is lower than the NRI value. The change in the method for calculating Kappa and NRI from a continuous risk to a unit change, i.e., unit-Kappa and unit-NRI, reduces the risk of obtaining a statistically significant result after adding a random variable to the logistic regression model.

## Figures and Tables

**Figure 1 ijerph-19-10213-f001:**
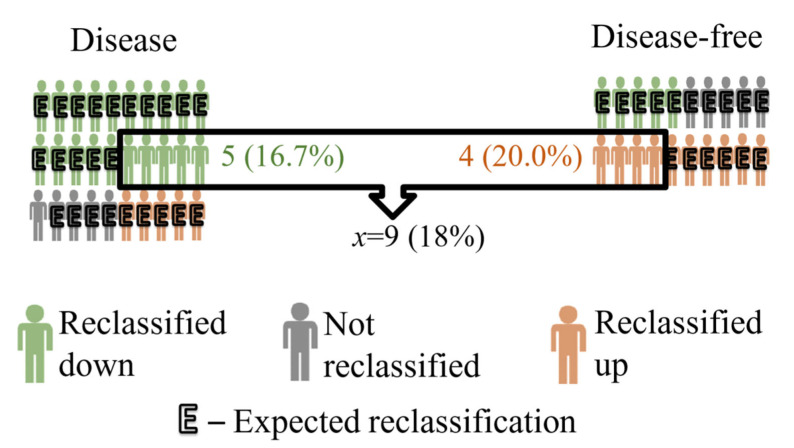
Method for presenting variable *x*, i.e., the number of correct reclassifications reduced by the random number of correct reclassifications after adding a new variable to the logistic regression model.

**Table 1 ijerph-19-10213-t001:** Observed reclassification to a group of people who were diagnosed with a disease and a group of disease-free individuals, and the numbers expected for random reclassification.

**Observed Frequency**				
		disease-free	diseased	total
reclassification	down	*a_O_*	*b_O_*	*# down*
	no changes	*c_O_*	*d_O_*	*# no changes*
	up	*e_O_*	*f_O_*	*# up*
	total	*# disease-free*	*# diseased*	*n*
**Expected frequency ***				
reclassification	down	*a_E_*	*b_E_*	
	no changes	*c_E_*	*d_E_*	
	up	*e_E_*	*f_E_*	

* expected frequencies were calculated in a standard way, i.e., by multiplying the marginal sums of rows and columns and dividing by sample size; # denotes the number of.

**Table 2 ijerph-19-10213-t002:** Location of the hidden category in the table with observed reclassification to the group of people who were diagnosed with the disease and the group of disease-free individuals, and table with expected numbers for a random reclassification.

**Observed Frequency**					
		disease-free	hidden category	diseased	total
reclassification	down	*a_O_*	0	*b_O_*	*# down*
	no changes	*c_O_*	0	*d_O_*	*# no changes*
	up	*e_O_*	0	*f_O_*	*# up*
	total	*# disease-free*	0	*# diseased*	*n*
**Expected frequency ***					
reclassification	down	*a_E_*	0	*b_E_*	
	no changes	*c_E_*	0	*d_E_*	
	up	*e_E_*	0	*f_E_*	

* expected frequencies were calculated in a standard way, i.e., by multiplying the marginal sums of rows and columns and dividing by the sample size; # denotes the number of.

**Table 3 ijerph-19-10213-t003:** Observed reclassification to the group of people who were diagnosed with a disease and the group of disease-free individuals, and the numbers expected random reclassification using a 50-element sample.

**Observed Frequency**					
		disease-free	hidden category	diseased	total
reclassification	down	*a_O_* = 20	0	*b_O_* = 5	*a_O_* + *b_O_* = 25
	no changes	*c_O_* = 5	0	*d_O_* = 5	*c_O_* + *d_O_* = 10
	up	*e_O_* = 5	0	*f_O_* = 10	*e_O_* + *f_O_* = 15
	total	*a_O_* + *c_O_* + *e_O_* = 30	0	*b_O_* + *d_O_* + *f_O_* = 20	*n* = 50
**Expected frequency ***					
reclassification	down	*a_E_* = 15	0	*b_E_* = 10	
	no changes	*c_E_* = 6	0	*d_E_* = 4	
	up	*e_E_* = 9	0	*f_E_* = 6	

* expected frequencies were calculated in a standard way, i.e., by multiplying the marginal sums of rows and columns and dividing by sample size.

**Table 4 ijerph-19-10213-t004:** Summary of results for one-dimensional logistic regression models describing CVD risk depending on BMI, place of residence, marital status, income, daily activity, education, SCORE result, and random variables with planned distributions.

Independent Variables	Frequency (%)	*p*-Value	OR [95%CI]	R^2^ #	BASIC MODEL *
**CANDIDATES FOR THE BASIC MODEL**					
**1. BMI**				0.02	**BMI**
underweight	21 (0.5)	0.2287	1.7 [0.72, 4.05]		
standard	931 (23.5)	reference			
overweight	1780 (44.8)	<0.0001	1.52 [1.29, 1.79]		
obesity	1239 (31.2)	<0.0001	1.97 [1.65, 2.35]		
**2. place of residence**				0.0003	
rural area	1496 (37.7)	0.3207	0.94 [0.82, 1.07]		
urban area	2475 (62.3)	reference			
**3. marital status**				0.0004	
single	1143 (28.8)	0.1202	0.9 [0.78, 1.03]		
in a relationship	2828 (71.2)	reference			
**4. income**				0.007	**income**
low	1034 (26.0)	0.0007	0.77 [0.67, 0.9]		
average	2226 (56.1)	reference			
high	711 (17.9)	0.0001	0.7 [0.59, 0.83]		
**5. daily activity**				0.003	**daily activity**
passive	1335 (33.6)	0.0035	1.29 [1.09, 1.53]		
mixed	1793 (43.8)	0.6809	0.97 [0.82, 1.14]		
active	897 (22.6)	reference			
**CANDIDATES FOR THE NEW MODELS**					
**ADDITIONAL**					
**6. education**				0.04	
basic	830 (20.9)	reference			
professional	1065 (26.8)	<0.0001	0.63 [0.52, 0.76]		
medium	1408 (35.5)	<0.0001	0.45 [0.38, 0.53]		
higher	668 (16.8)	<0.0001	0.40 [0.33, 0.49]		
**7. SCORE**				0.39	
high	2573 (64.8)	<0.0001	22.66 [18.27, 28.12]		
low	1398 (35.2)	reference			
**RANDOM**	assumed parameters				
**8. uniform**	interval: [0, 100]	0.3049	1.00 [1.00, 1.00]	0.0003	
**9. normal**	mean (sd) = 0 (1)	0.4043	1.03 [0.96, 1.09]	0.0003	
**10. Poisson**	λ = 4	0.0443	1.03 [1.00, 1.07]	0.001	
**11. exponential**	λ = 1	0.5114	1.02 [0.96, 1.09]	0.0001	
**12. binomial**	*p* = 0.1	0.7362	0.96 [0.78, 1.19]	0.00003	
**13. binomial**	*p* = 0.5	0.6574	0.97 [0.86, 1.10]	0.00009	

OR [95%CI]—Odds Ratio with 95% Confidence Interval. *p*-value of the Wald test. * variables remaining in basic model based on forward stepwise regression. # R^2^ (Nagelkerke)—the measure of the model fit quality.

**Table 5 ijerph-19-10213-t005:** Results of logistic regression models describing CVD risk: basic model (variables in the model: BMI, income, and daily activity) and models expanded by education, SCORE, and random variables with uniform, normal, Poisson, exponential, binomial (*p* = 0.1), and binomial (*p* = 0.5) distributions.

Model	Wald Test *p*-Value	Likelihood Ratio Test *p*-Value	AUC [95%CI]	AUC Change after Adding Marker *p*-Value
basic			0.59 [0.57, 0.60]	
basic + education	(*p* < 0.0001 for each category)	<0.0001	0.63 [0.61, 0.65]	<0.0001
basic + SCORE	<0.0001	<0.0001	0.79 [0.78, 0.81]	<0.0001
basic + uniform	0.2532	0.2532	0.59 [0.57, 0.61]	0.3989
basic + normal	0.5251	0.5251	0.59 [0.57, 0.60]	0.7074
basic + Poisson	0.0550	0.0549	0.59 [0.57, 0.61]	0.3206
basic + exponential	0.4761	0.4764	0.59 [0.57, 0.61]	0.4795
basic + binomial (*p* = 0.1)	0.7848	0.7847	0.59 [0.57, 0.60]	0.4742
basic + binomial (*p* = 0.5)	0.8866	0.8866	0.59 [0.57, 0.60]	0.6523

**Table 6 ijerph-19-10213-t006:** Reclassification quality based on Cohen Kappa and Net Reclassification Improvement (NRI) for a continuous change of CVD risk between the base and extended logistic regression models.

Model	*x* Number (% from *n*)	*p*-Value *	κ [95%CI]	*p*-Value #	NRI [95%CI]
basic + education	311 (7.82)	<0.0001	0.16 [0.13, 0.19]	<0.0001	0.32 [0.26, 0.38]
basic + SCORE	1035 (26.06)	<0.0001	0.50 [0.48, 0.53]	<0.0001	1.06 [1.01, 1.10]
basic + uniform	30 (0.74)	0.3470	0.01 [−0.02, 0.05]	0.3470	0.03 [−0.03, 0.09]
basic + normal	17 (0.41)	0.6068	0.01 [−0.02, 0.04]	0.6068	0.02 [−0.05, 0.08]
basic + Poisson	54 (1.35)	0.0876	0.03 [0.00, 0.06]	0.0874	0.05 [−0.01, 0.12]
basic + exponential	−22 (−0.55)	0.4733	−0.01 [−0.04, 0.02]	0.4736	0.02 [−0.04, 0.08]
basic + binomial (*p* = 0.1)	0 (0.00)	0.6690	0.00 [−0.02, 0.02]	0.6684	0.01 [−0.03, 0.05]
basic + binomial (*p* = 0.5)	14 (0.35)	0.6574	0.01 [−0.02, 0.04]	0.6574	0.01 [−0.05, 0.08]

*x*–number: the number of correct reclassifications reduced by the random number of correct reclassifications. κ [95%CI]: Kappa coefficient of agreement with 95% Confidence Interval. * testing the significance of the Kappa coefficient. NRI [95%CI]: Net Reclassification Improvement with 95% Confidence Interval. # testing the significance of the NRI.

**Table 7 ijerph-19-10213-t007:** Reclassification quality based on Cohen Kappa and Net Reclassification Improvement (NRI) for a change of CVD risk (between the base and extended logistic regression models) exceeding 1%.

Model	*x* Number (% from *n*)	*p*-Value *	Unit-κ [95%CI]	*p*-Value #	Unit-NRI [95%CI]
basic + education	310 (7.8)	<0.0001	0.15 [0.12, 0.18]	<0.0001	0.31 [0.26, 0.38]
basic + SCORE	1035 (26.1)	<0.0001	0.50 [0.48, 0.53]	<0.0001	1.05 [1.01, 1.10]
basic + uniform	24 (0.6)	0.1965	0.007 [−0.004, 0.018]	0.1984	0.02 [−0.01, 0.06]
basic + normal	−6 (−0.2)	0.3595	−0.002 [−0.005, 0.002]	0.3599	−0.006 [−0.019, 0.007]
basic + Poisson	30 (0.7)	0.1588	0.010 [−0.004, 0.023]	0.1590	0.030 [−0.011, 0.072]
basic + exponential	8 (0.2)	0.2918	0.002 [−0.002, 0.006]	0.2950	0.008 [−0.07, 0.023]
basic + binomial (*p* = 0.1)	0 (0)	1.0000	0.000 [0.000, 0.000]	NA	0.000 [0.000, 0.000]
basic + binomial (*p* = 0.5)	0 (0)	1.0000	0.000 [0.000, 0.000]	NA	0.000 [0.000, 0.000]

*x*–number: the number of correct reclassifications reduced by the random number of correct reclassifications, determined for the unit probability change. unit-κ [95%CI]: Kappa coefficient of agreement with 95% Confidence Interval for unit probability change. * testing the significance of the Kappa coefficient. unit-NRI [95%CI]: Net Reclassification Improvement with 95% Confidence Interval for unit probability change. # testing the significance of the NRI.

**Table 8 ijerph-19-10213-t008:** Reclassification quality based on Cohen Kappa and Net Reclassification Improvement (NRI) for a categorial risk of the CVD (between the base and extended logistic regression models). The two risk categories were built based on a cut-off value *p* = 0.4444.

Model	*x* Number (% from *n*)	*p*-Value *	κ (*p*) [95%CI]	*p*-Value #	NRI (*p*) [95%CI]
basic + education	52 (1.3)	0.0012	0.01 [0.01, 0.02]	<0.0001	0.06 [0.03, 0.09]
basic + SCORE	397 (10.0)	<0.0001	0.13 [0.11. 0.14]	<0.0001	0.40 [0.37. 0.44]
basic + uniform	−6 (−0.2)	0.3936	−0.002 [−0.001, 0,002]	0.3934	−0.007 [−0.022, 0.009]
basic + normal	2 (0.1)	0.3332	0.001 [−0.001, 0.002]	0.2749	0.004 [−0.003, 0.011]
basic + Poisson	9 (0.2)	0.2882	0.002 [−0.002, 0.007]	0.2879	0.009 [−0.008, 0.026]
basic + exponential	4 (0.1)	0.1987	0.001 [−0.001, 0.002]	0.1999	0.004 [−0.002, 0.010]
basic + binomial (*p* = 0.1)	0 (0.0)	0.9092	0.000 [−0.002, 0.002]	0.9998	0.000 [−0.004, 0.004]
basic + binomial (*p* = 0.5)	0 (0.0)	1.0000	0.000 [0.000, 0.000]	NA	0.000 [0.000, 0.000]

*x*–number: the number of correct reclassifications reduced by the random number of correct reclassifications, determined for the probability cut-off. unit-κ [95%CI]: Kappa coefficient of agreement with 95% Confidence Interval for the probability cut-off. * testing the significance of the Kappa coefficient. unit-NRI [95%CI]: Net Reclassification Improvement with 95% Confidence Interval for the probability cut-off. # testing the significance of the NRI. NA—not available.

## Data Availability

Data available on request due to restrictions privacy and ethical.

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
