# Peer review of "Cohen’s Kappa Coefficient as a Measure to Assess Classification Improvement following the Addition of a New Marker to a Regression Model"

_ijerph, 2022, doi:10.3390/ijerph191610213_

Round 1
Reviewer 1 Report
The article presents a very fashionable topic, which comes from the origin of the methods of statistical inference by assuming the p-value as a parameter of interpretation.
The relationship between p-values and inference on hypotheses is a critical point since the interpretation of statistical analysis depends on it. Since 2017 it is one of the key topics of the American Statistical Association statement on statistical significance and p-values [1], therefore studies comparing other indicators to measure the significance and robustness of the hypotheses posed help to improve in the decision making process.
As Cohen [2] indicates, the difference in weight between an 18-year-old girl and a 30-year-old girl, a 40-year-old girl or a 60-year-old girl may be significant, but Cohen's d will be different, less than 0.2, less than 0.5, less than 0.8 or greater than this value. Therefore, this parameter, in addition to providing a value for significance, indicates the strength of the relationship.
The bibliography could be further updated: the first reference is from 2019 and then 2 from 2017, the rest of the references are prior to those dates.
[1] Altman, N., Krzywinski, M. Interpreting P-values. Nat Methods 14, 213-214 (2017). https://doi.org/10.1038/nmeth.4210
[2] J. Cohen, Statistical power analysis for the behavioral sciences, Lawrence Erlbaum Associates, Publishers, 1988.
Author Response
Dear Sir/Madam
Thank you very much for your review and valuable comments. I have modified the content of the manuscript to implement your suggestions.
The Kappa coefficient we used is one measure of effect size. The statistical community has for many years emphasized the need to report effect sizes instead of focusing solely on the p-values of statistical tests. Therefore, paying attention to this aspect fits into the context of the paper and improves its understanding.
In the introduction of the paper [line from 33 to 39], we added a passage describing the necessity of reporting effect sizes. We cited two papers suggested by the reviewer and six more showing how important this aspect is in the various fields of science (the papers we cited are in the fields of psychology, education and medicine).

Reviewer 2 Report
In current study, it is to use extend the Cohen Kappa coefficient to check the reclassification quality in a regression model. It is a valuable study and provide a measures for this reserch topic. There are some comments, I hope author could answer.
1. In table4, the classfication criterion of variables should be provided more information.
2. In table 6, the meaning of "x number (%from n) p-value κ [95%CI] p-value NRI [95%CI]" ; In table 7, the meaning of "x number (%from n) p-value unit-κ [95%CI] p-value unit-NRI [95%CI]". please add some information and calculation under the table. It will be better for reading. In table 8, please add some information too.
3. The reason of these models" basic, basic + education, basic + SCORE; basic + uniform, basic + normal, basic + Poisson, basic + exponential, basic + binomial (p=0.1), basic + binomial (p=0.5)" Why the basic model did not includ gender, age, or the history of diseases? Why did not consider basic + education+SCORE? and Why did not consider the models, such as basic + education + Poisson/ exponential/ binomial (p=0.1)/ binomial (p=0.5)?
4. There are some citation format in the text should be corrected.
Author Response
Dear Sir/Madam
Thank you very much for your review.
I consider your remarks very valuable. I have analysed them and modified the content of the manuscript to implement them. I believe that thanks to the advice provided, the work becomes clearer and more valuable for the readers.
Please find my detailed replies to the passages of the article you have indicated. The changes were also introduced in the paper.
- Regarding comment 1 and comment 3:
What was indeed missing in the paper was a clear presentation of how variables were selected for the base model and extended models. The purpose of our work was not to propose yet another CVD prediction model, so we did not want to select a base model in the best way and then expand with more variables, which is very often the final goal of building prediction models. The priority was to select variables that would extend the model in such a way as to improve its prediction (hence the selection of the two best variables, with the largest R2) and one that would not improve the prediction (hence the random variables added to the model) to show how the Kappa coefficient works. The way the base model was built was not a priority in our approach, it was built later, based on the variables remaining after selection for the extended models
The detailed description of the approach outlined above, we have tried to present in a way that answers exactly the questions posed by the reviewer - this can be found in the Plan of simulation study section. We have also added additional information to Table 4 to explain which variables are in the base model and why. Above Table 4 we have added two additional sentences drawing the reader's attention to the issues of selecting variables for each model.
- Regarding comment 2:
We have added relevant explanations beneath the tables
- Regarding comment 4:
We have standardized the format of citations

Round 2
Reviewer 2 Report
Thank you for your reply. It had addressed my comments.